# Development of an Improved Method for the Determination of Iodine/β-Cyclodextrin by Means of HPLC-UV: Validation and the Thyroid-Stimulating Activity Revealed by In Vivo Studies

**DOI:** 10.3390/pharmaceutics13070955

**Published:** 2021-06-25

**Authors:** Avez Sharipov, Zufar Boboev, Sunnatullo Fazliev, Shokhid Gulyamov, Akhmatkhodja Yunuskhodjayev, Jamoliddin Razzokov

**Affiliations:** 1Department of Inorganic, Physical and Colloidal Chemistry, Tashkent Pharmaceutical Institute, Oybek Street 45, Tashkent 100015, Uzbekistan; sharipov_a@pharmi.uz (A.S.); boboev_z@pharmi.uz (Z.B.); shohid9395@chungbuk.ac.kr (S.G.); 2Medical Devices and Medical Equipment, State Centre of Expertize and Standardization of Medicines, Tashkent 100002, Uzbekistan; 3Max Planck School Matter to Life, Jahnstraße 29, 69120 Heidelberg, Germany; sunnatullo.fazliev@mtl.maxplanckschools.de; 4Faculty of Chemistry and Earth Sciences, Heidelberg University, Im Neuenheimer Feld 234, 69120 Heidelberg, Germany; 5College of Pharmacy, Chungbuk National University, 194-21, Osongsaengmyeong 1-ro, OSONG-Eup, Heungdeok-gu, Cheongju City 28160, Chungbuk, Korea; 6The Innovation Center of Pharmacy, Tashkent 100015, Uzbekistan; akhmad5604@yandex.ru; 7Department of Physics and Chemistry, Tashkent Institute of Irrigation and Agricultural Mechanization Engineers, Kori Niyoziy 39, Tashkent 100000, Uzbekistan; 8Department of Physics, National University of Uzbekistan, Tashkent 100174, Uzbekistan; 9Institute of Material Sciences, Academy of Sciences, Chingiz Aytmatov 2b, Tashkent 100084, Uzbekistan; 10College of Engineering, Akfa University, Kichik Halqa Yuli Street 17, Tashkent 100095, Uzbekistan

**Keywords:** iodine, potassium iodide, iodine deficiency, β-cyclodextrin, iodine/β-CD, thyroid-stimulating activity, methimazole

## Abstract

Iodine, being an intrinsic part of thyroid hormones, is a vital microelement required for normal growth and development, particularly in children. Inadequate daily intake of iodine causes iodine deficiency, which is responsible for several health disorders, such as cretinism and goiters. Therefore, the development of new drugs and/or food supplements for iodine deficiency is crucial. We synthesized an iodine/β-cyclodextrin complex based on a host–guest model, and in this paper, we outline the development of a new quantitative analysis method. We suggest a robust and reliable high-performance liquid chromatography method to determine the total amount of iodine species in the complex. Moreover, we performed validation of our method. The results of validation presented here show the reliability, accuracy and high precision of the method. Furthermore, for the first time, we show results of in vivo studies for the thyroid-stimulating activity of the iodine/β-cyclodextrin complex. Our findings indicate that the thyroid-stimulating activity of iodine/β-cyclodextrin is comparable to that of potassium iodide, which is the main active pharmaceutical substance of conventional drugs for iodine deficiency.

## 1. Introduction

Iodine, as a microelement, is required for normal growth and development. In human organisms, iodine is an indispensable part of thyroid hormones. Iodine deficiency as a global public health problem affects around 2 billion people in the world [1]. The majority of investigations have shown that iodine deficiency primarily causes mental development disorders in young generations [2]. Specifically, insufficient intake of iodine leads to poor school performance and cretinism, which is associated with impaired cognitive development [3,4,5]. To prevent the similar consequences, it is recommended to consume 120 mg iodine for children, 250 mg for pregnant women and 150 mg for adults per day [6]. Thus, adequate dietary intake of iodine is critical to prevent iodine deficiency. To solve this issue, potassium and sodium iodides are commonly used in the food and pharmaceutical industries. It is known that iodides play a vital role in the production of iodine-containing hormones by the thyroid gland [7]. Although many food supplements and drugs are available for iodine deficiency, there still remain some challenges. In particular, it is crucial to prevent the oxidation of iodides and to improve the solubility of iodine in physiological conditions. There is plenty of research on record to address such problems; for instance, it has been shown that α-cyclodextrin forms a complex with a mixture of potassium iodide and iodine [8].

Cyclodextrins (CDs) are important oligosaccharides and are produced via enzymatic hydrolysis of amylose. CDs were extensively used in the early years of supramolecular chemistry due to their inexpensive price and high accessibility. They maintain a central cavity with a certain size depending on the number of glucoses in the cycle [9]. The central cavity as a functional site of CD is involved in transportation of molecules and is widely applied in drug delivery [10]. When the size and shape of guest molecules match with those of the host (CD), substantial binding affinities can be observed [11,12,13,14,15]. Here, physical and chemical characterization of host–guest interactions is crucial, since this enables the exploitation of the full potential of host–guest complex molecules. Various methods are used to synthesize host–guest complexes in solutions and in solid phases [16,17]. Among these methods, liquid chromatography has been extensively used to study CDs and their complexes with iodine [18]. The host–guest interactions can lead to enhanced physical and biological properties of the guest substance in physiological conditions; therefore, CDs have lately been a constant research focus in pharmaceutics [19,20,21].

β-CD was observed to form a more stable complex with iodine than α-CD, and this kind of complex formation in general increases the solubility of iodine in water [22]. The size of the internal cavities of α-CD and β-CD were found to be 4.7–5.3 Å and 6.0–6.5 Å, respectively [16]. The determined diameter of an iodine molecule is 4.94 Å [16,23]. Thus, the stable complex formation of β-CD with iodine molecules is comparatively more probable than of α-CD [24,25]. Several similar host–guest complexes of iodine/β-CD have already been synthesized, and their potential pharmacological activities also have been determined. For example, Wang et al. showed the bacteriostatic and antifungal activities of iodine/β-CD [26]. Moreover, a recent study revealed that the consumption of sausages fortified with iodine/β-CD complex positively affected the iodine status of volunteers [27].

Regarding the quality control of iodine/β-CD complexes, the iodimetric titration method is usually used for quantitative analysis purposes [26,27]. However, the total amount of iodine species in iodine/β-CD complex has not been quantified hitherto using the high-performance liquid chromatography (HPLC) method. In this study, we present a robust and reliable HPLC method to quantify the total amount of iodine in a iodine/β-CD complex that we previously synthesized [28]. This complex contains not only iodine, but iodide ions as well, and we show that our novel HPLC analysis method offers a simple and accurate way of determining the total amount of iodine in the complex. In addition, we investigated the formation of the complex and validated the HPLC method. Further, we report results of in vivo study of the thyroid-stimulating activity of iodine/β-CD complex in comparison with that of potassium iodide. 

## 2. Materials and Methods

### 2.1. Materials

I_2_, β-cyclodextrin (β-CD), KI, Na_2_S_2_O_3_ and KH_2_PO_4_ were purchased from Sigma Aldrich. All the chemicals were analytical reagent grade. Acetonitrile and methanol (for HPLC—ISOCRATIC GRADE) were from VWR International S.A.S (Fontenay-sous-Bois, France). Tetramethylammonium hydroxide (TMAH) 25% (*m/m*) in water was from Merck (Darmstadt, Germany), Methimazole (MMI) (Sigma-Aldrich, St. Louis, MO, USA).

Deionized water, resistivity 18 MΩcm, obtained from a SARTORIUS water-purification system (ARIUM Comfort 1 UV, Sartorius Weighing Technology GmbH, Göttingen, Germany), was used throughout this work. 

### 2.2. Synthesis of Iodine–β-Cyclodextrin Complex

Synthesis of the iodine/β-CD complex was carried out according to the synthesis procedure described by Wang et al. [26]. Briefly, 3.8 g of KI was dissolved in 15 mL of deionized water. Consequently, the prepared solution was mixed with 0.38 g of I_2_ (1.5 mM) and was left 10 min to form polyiodide. Next, this solution was dropped into 30 mL β-cyclodextrin solution (0.669 g, 1.5 mM) contained in a 100 mL conical flask. The mixture was stirred for 5 h at 25 °C and then retained for 10 h at 4 °C in order to fully encapsulate iodine in β-CD. After that, the brown powder of iodine/β-CD was filtered, washed with KI solution to remove I_2_ molecules from the filter cake and rinsed with deionized water to get rid of both KI and β-CD. The final product, iodine/β-CD, was dried in vacuum for 4 h at 40 °C. Yield—81.4%, humidity—2.4%, purity—97.32%.

### 2.3. Powder X-ray Diffraction (XRD) Investigation

XRD patterns were recorded using an XRD-6100 X-ray diffractometer (Shimadzu, Kyoto, Japan) at 30 kV, 30 mA (CuKα monochromatic radiation, λ(CuKα) = 1.54178 Å), step size 0.02° 2θ, counting time 6 s per step, over the angular range 4–80° 2θ.

### 2.4. HPLC Analysis

High-performance liquid chromatograph system–Shimadzu LC-20 Nexera XR, solvent supply system—LC-20AD, degasser—DGU-20A5R, autosampler—SIL-20A, detector—SPD-M20A (Diode array detector), column thermostat—CTO-20A from (Shimadzu Europa GmbH, Duisburg, Germany); column—ZORBAX Eclipse Plus, C18, 150 × 4.6 mm, 3.5 µm, from Agilent (Santa Clara, CA, USA).

The mobile phase, consisting of a mixture of water phase (containing 67 mmol·L^−1^ potassium dihydrogen phosphate) and acetonitrile in ratio 75:25 (*v*/*v*, each liter of mobile phase containing 1.7 g tetramethylammonium sulfate) was delivered isocratically at a flow rate of 1.0 mL·min^−1^. Following its preparation, the mobile phase was filtered under vacuum through a 0.45 μm membrane filter and ultrasonically degassed prior to use. The chromatographic separation was performed on a ZORBAX Eclipse Plus, C18, 150 × 4.6 mm, 3.5 µm, from Agilent (Santa Clara, CA, USA). The column temperature was maintained constantly at 40 °C using a thermostatically controlled column oven. UV-Vis spectrophotometric detection was performed at 237 nm wavelength. The chromatographic running time for each analysis was 5.0 min.

### 2.5. Preparation of Solutions

#### 2.5.1. Preparation of Standard Solutions of KI

The stock solution of iodide at a concentration of 1 mg/mL was prepared by dissolving KI (13.08 mg) in 10 mL deionized water. From this stock solution 20, 40, 60, 80, 100, 120, 140 μg/mL standard solutions of iodide were prepared.

#### 2.5.2. Preparation of Sample Solution

Sample solution with a 0.1 mg/mL concentration of iodine/β-CD was prepared. Briefly, weighted 10 mg of iodine/β-CD was dissolved in 50 mL deionized water in a 100 mL volumetric flask. The solution was left in ultrasonic bath for 15 min. Then, the flask was filled with de-ionized water and filtered through a filter with pore size of 0.45 μm.

#### 2.5.3. Preparation of Iodine/β-CD Sample Solution to Quantify Total Amount of Iodine

To determine total amount of iodine in iodine/β-CD complex, 10 mg of the complex was dissolved in 50 mL water in 100 mL volumetric flask and 0.04 M solution of sodium thiosulfate was added until the solution became colorless. Then, the flask was filled with deionized water and filtered through a filter with pore size of 0.45 μm.

#### 2.5.4. Preparation of 0.04 M Solution of Sodium Thiosulfate

0.6205 g of sodium thiosulfate pentahydrate was dissolved in deionized water in 100 mL volumetric flask.

### 2.6. Thyroid-Stimulating Activity

#### 2.6.1. Animals

All the experimental procedures were performed in accordance with the regulations of good laboratory practice, approved by the State Center for Expertise and Standardization of Medicines, Medical Devices and Medical Equipment under the Ministry of Health of the Republic of Uzbekistan. Seven-week-old Wistar rats were purchased. Animals were housed in standard cages under controlled temperature, humidity and light, were given standard food and had access to water. Animals could acclimatize to the conditions of the animal facility for a week.

#### 2.6.2. Experimental Design

The thyroid-stimulating activity study was performed by using a methimazole hypothyroidism model [29,30]. After one week of adaptation, 8-week-old Wistar rats were divided into 4 groups, with 6 rats in each. All groups received drugs and solvent at a volume of 2 mL per 200 g body weight. In the control group, deionized water was administered by oral gavage to the rats. Animals of the hypothyroid group received MMI at a dose of 20 mg/kg, solved in deionized water daily for 10 days. Iodine–β-CD was administered to rats in the test group at 1 mg/kg body weight daily for 10 days before an hour of the administration of MMI. In the comparison group, rats received potassium iodide at a dose of 1 mg/kg daily for 10 days before an hour of the administration of MMI. All animals were anesthetized on the 11th day of the experiment by intraperitoneal injection of 1.2% avertin solution. 

#### 2.6.3. Measurement of Serum TSH, T3 and T4

Blood samples were collected via cardiac puncture and put into tubes with an anticoagulant. They were centrifuged in a centrifuge at 3000 rpm for 5 min. The serum of each blood sample was put into another tube and was kept at −80 °C until further analysis. TSH, T3 and T4 levels in the serum were measured by ELISA kits and read in MR-96A microplate reader, Mindray(Darmstadt, Germany).

### 2.7. Statistical Analysis

All the experiments were executed in triplicate, and the results are presented with mean ± standard error (SE). The descriptive statistics, Student’s *t*-test, analysis of variance (ANOVA), line diagrams and Duncan’s multiple range test (DMRT) were made using Excel 2010 and SPSS (Ver 2016, IBM, Armonk, NY, USA). The difference of *p* < 0.05 was considered as statistically significant among the factors. Analysis of linearity and correlation coefficient was performed using GraphPad Prism 9 program.

## 3. Results and Discussion

### 3.1. Results of XRD Analysis

XRD analysis is one of the common methods to study complex formation [31]. We carried out XRD analysis of the starting materials: iodine, potassium iodide and β-CD, their stoichiometric mixture in a mass ratio of 1:1 and obtained the complex compound (Figure 1).

In the XRD spectrum of the stoichiometric mixture of iodine with potassium iodide, we observed peaks with strong intensities at 22.02, 25.46 and 36.20 2theta values (Figure 1I). The same peaks can be observed in the XRD spectrum of iodine/β-CD complex, but with substantially lower intensities. The spectrum of the mixture of iodine, potassium iodide and β-CD shows many unclear peaks, which mostly overlap with each other. Nevertheless, three distinct peaks at 18.31, 18.96 and 21.55 2theta values can be identified (Figure 1II). The XRD spectrum of the host molecule is characterized by many peaks; especially strong peaks at 11.95, 14.53, 15.56, 17.62, 18.67, 20.77 and 23.92 2theta values can be seen (Figure 1III). 

The spectrum of iodine/β-CD lacks several characteristic peaks, which are present in the spectra of the starting materials, but rather shows three main peaks at 14.88, 17.94 and 19.19 2theta values, while a peak at 19.19 2theta is defined as a characteristic peak for the iodine/β-CD complex (Figure 1IV). Overall, XRD spectra of the starting materials, their stoichiometric mixture and the complex show clear differences and patterns that are unique for each compound and therefore prove the formation of iodine/β-CD complex.

### 3.2. Results of HPLC Analysis

#### 3.2.1. Development of HPLC Method to Quantify Total Amount of Iodine in Iodine/β-CD Complex

The choice of solvent is crucial in our study. On the one hand, the solvent should facilitate release of the guest from the complex and dissolve the guest molecules. To transfer iodine from the complex, we reduced it to iodide, which is soluble in water and does not stay in the cavity of the host. On the other hand, in HPLC methods, components of the mobile phase have to mix well, dissolve the analyte and not cause any problems with the chromatographic column. In various analysis methods of iodine/β-CD, dimethylformamide and methanol–water systems were used [26]. In our method, we used water, methanol and water–methanol (40:60) systems as solvents. Since all three solvent systems showed very similar results, in further analysiswe used water as a solvent. 

In the cavity of the host, molecular iodine is expected to be in the form of complex compound with potassium iodide. From the synthesis and the yield of the complex, we speculated that iodine species are mostly in the form of potassium polyiodide [25]. To transfer all iodine from the complex to the solution, we used sodium thiosulfate as a reducing agent. Sodium thiosulfate reduces iodine to iodide (Reaction (1)), which is soluble in water and does not interact with the host.

Reaction (1):(1)$$I_{2}+2S_{2}O_{3}^{2-}\to 2I^{-}+S_{4}O_{6}^{2-}$$

In order to study the effect of sodium thiosulfate and its oxidized form (tetrathionate) in the separation process, we ran chromatographic analysis of the potassium iodide solution (Figure 2I), the sodium thiosulfate solution (Figure 2II) and the solutions resulting from reduction of the iodine/β-CD sample with excess (Figure 2III) and equivalent amounts of thiosulfate (Figure 2IV).

As can be seen from the chromatograms (Figure 2), the retention times of the reducing agent (1.78 min) and its oxidized form (2.82 min) are quite different from that of the analyte (2.12 min). Chromatograms do not show any overlap of peaks, thus proving the chosen conditions to be well developed. 

#### 3.2.2. Results of HPLC Analysis to Determine Total Amount of Iodine in the Complex

The total amount of iodine was determined via HPLC analysis (see Section 2.4). Retention time of iodide released from the complex was 2.12 min (Figure 2IV). First, we determined the amount of iodide ions released from the complex. Then, using thiosulfate as a reducing agent, we transferred the molecular iodine to iodide and quantified the total amount of iodine species as iodide (Table 1). We found that iodine/β-CD complex contains 7.88 ± 0.2% (*n* = 25) iodide ions and 19.91 ± 0.3% iodine species in total. Wang and coworkers previously determined the amount of iodine in iodine/β-CD complex using iodimetric titration and reported that the total amount of iodine was 17.32%, indicating 1:1 ratio of iodine:β-CD in the complex [26]. M. Polumbryk et al. also synthesized iodine/β-CD and found that the synthesized complex contains 16.82 ± 0.4% iodine in total [27]. The complex synthesized by us contains 2.59–3.09% more iodine in total compared to the abovementioned complexes. We believe that this increase is either due to additional encapsulation of potassium iodide by the host or by the formation of more complex polyiodides during synthesis.

### 3.3. Studying the Yield of the Transfer of Iodine Species to Iodide

The quantitative transfer of iodine species to iodide is important for the analysis, since incomplete reduction is a source of intrinsic error. Thus, we investigated the yield of the reduction of iodine species with thiosulfate. To accomplish this task, we mixed equivalent amounts of molecular iodine and sodium thiosulfate and ran chromatographic analysis of the resulting solution. We also performed additional chromatographic analysis of the solution after mixing molecular iodine with an excess amount of sodium thiosulfate. 

An amount of 10 mL of the 0.05 M solution of molecular iodine was added to 100 mL volumetric flask, and then 20 mL deionized water was added. Next, the 0.1 M solution of sodium thiosulfate was added until the solution became colorless and the flask was filled with deionized water. Then, a 5 mL aliquot of the resulting solution was added to the 50 mL volumetric flask and filled with deionized water to get a new solution of iodide with approximately 0.127 mg/mL concentration. We performed HPLC analysis of this new iodide solution. The results of the analysis with yield of the reduction reaction are given in Table 2. The results show that the yield of transfer of iodine to iodide is 99.564 ± 0.27%. Such high transfer yield significantly reduces error in the analysis.

### 3.4. Method Validation

The method was validated in accordance with the guidelines on bioanalytical method validation from the European Medicines Agency (EMA, 2011) [32,33]. Specificity, linearity, limit of detection (LOD), limit of quantification (LOQ) and accuracy were verified.

#### 3.4.1. Specificity

The specificity of the method was tested by investigating effects of the mobile phase, the solvent and other components of the matrix on detection of the analyte. The close retention times of the standard solution of iodine (2.080 min) and of the sample solution (2.097 min) show that the method is very specific.

#### 3.4.2. Linearity

To study linearity of our new analytical method, we ran chromatographic analysis of standard solutions of KI with concentrations ranging from 20 to 140 mg/mL, five times for each concentration. The results of the experiment were analyzed with GraphPad Prism 9 program. HPLC analyses performed at different concentrations of potassium iodide show performance and linear regression with a correlation coefficient greater than r^2^ > 0.9989 (Table 3, Figure 3). Potassium iodide concentrations were determined by interpolating intensity ratios corresponding to calibration curves. 

#### 3.4.3. Limit of Detection and Limit of Quantitation

The limit of detection (LOD) was determined as a statistical concentration of iodine within a confidence interval of 99%. HPLC analysis of 10 µg/mL solutions of KI were performed seven times and an average of the results was calculated. Average standard deviation was determined, and by multiplying it by Student’s t-value of 3.143, LOD was found. The value of LOD was 1.24 µg/mL for KI. The value for the limit of quantification (LOQ) (4.14 µg/mL) turned out to be three-fold of LOD and corresponds to the lowest concentration in the calibration graph (Table 4).

Repeated analysis (*n* = 5) results showed 98.41–102.87% repeatability compared to calculations, with an RSD value of 1.33%.

#### 3.4.4. Accuracy

Accuracy is the closeness of agreement between the accepted true value, or a reference value, and the measured result obtained. Accuracy studies are usually evaluated by determining the recovery of a spiked sample of the analyte into the matrix of the sample to be analyzed [34]. Three different samples of the working solution of KI (low, medium and high, i.e., 80%, 100% and 120%, respectively) were analyzed (Table 4). 

### 3.5. Results of Thyroid-Stimulating Activity and Discussions

The results of the control, hypothyroid, test and comparison groups were compared and are shown in Figure 4. The thyroid gland releases the two main thyroid hormones, triiodothyronine (T3) and thyroxine (T4). T3 is the active form of thyroid hormone, and T4 equals more than 80% of the secreted hormone. TSH stimulates thyroid follicular cells to release thyroid hormones in the form of T3 or T4. It is well known that with low levels of TSH, the lack of stimulation of thyroid follicular cells causes a reduction in T3 and T4 levels, which leads to hypothyroidism [35]. For this reason, we chose these hormones. As can be seen from Figure 4a, there was a considerable reduction in the case of triiodothyronine (T3) in the hypothyroid group in comparison with the control group, the former of which showed a 4.3-fold decrease and indicated that hypothyroidism was induced successfully. Statistically significant increments in the level of serum T3 were observed in the test and comparison groups in comparison with the hypothyroid group, 2.3 and 1.8-fold higher, respectively. Furthermore, iodine/β-CD significantly increased the level of T3 when compared with the group that received potassium iodide. According to the results shown in Figure 4b, it was found that iodine/β-CD and potassium iodide increased the production of serum T4 hormone almost to the same level. As shown in Figure 4c, serum TSH level was elevated by the administration of ioodine/β-CD and potassium iodide, which showed nearly equal amounts, 0.041 and 0.038 mLE/L, respectively. However, TSH was not found in the control and hypothyroid groups. A significant difference was not observed between the test and comparison groups regarding all types of hormones, as determined by the ELISA kits and read in MR-96A microplate reader.

The amounts of serum T3 and T4 significantly diminished in the hypothyroid group in comparison with the control group. Iodine/β-CD demonstrated statistically significant increases in the levels of T3, T4 and TSH in the blood serum of the experimental animals in the model of methimazole hypothyroidism. When compared the results of the thyroid-stimulating activity of the iodine/β-CD and potassium iodide, it was found that the levels of T3, T4 and TSH in the blood serum showed very similar results.

## 4. Conclusions

In summary, this study suggests an improved method of HPLC analysis of iodine/β-CD to determine the iodide content and total iodine separately. It was also found that the iodine content in iodine/β-CD was 19.91 ± 0.3%. This is 2.59–3.09% more in comparison with previous studies [26,27]. The accuracy of the HPLC method was validated in terms of specificity, linearity, accuracy, limit of detection and limit of quantification. The thyroid-stimulating activity of iodine/β-CD was tested in rats using the methimazole hypothyroidism model. The thyroid-stimulating activity of iodine/β-CD at a dose of 1 mg/kg has been demonstrated in in vivo experiments. Our results showed slightly higher thyroid-stimulating activity of iodine/β-CD compared to potassium iodide. The results indicate that iodine/β-CD can be a potential drug for iodine deficiency. In the future, our research will focus on developing a new drug based on iodine/β-CD and conducting respective pre-clinical and clinical trials.

## Figures and Tables

**Figure 1 pharmaceutics-13-00955-f001:**
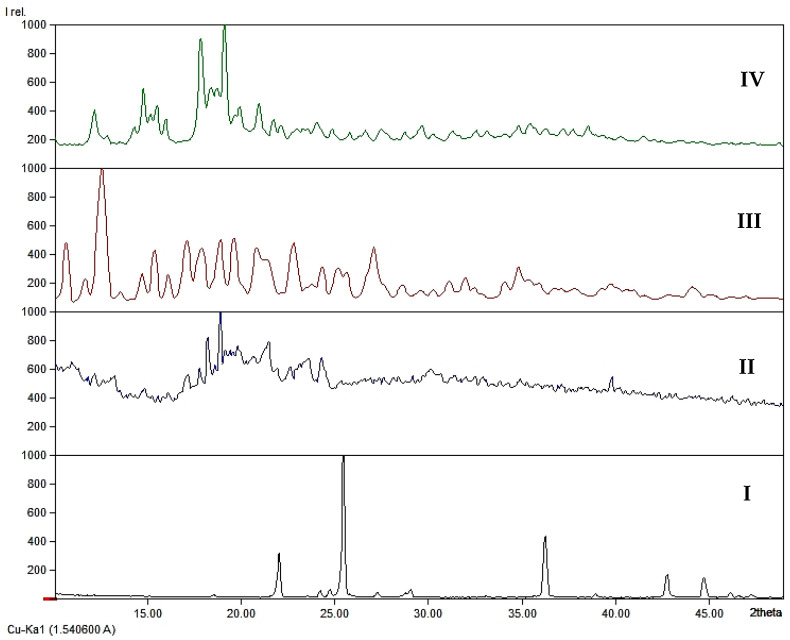
XRD spectra for (**I**) the stoichiometric mixture of KI and I_2_; (**II**) the stoichiometric mixture of KI, I_2_ and β-CD; (**III**) β-CD; (**IV**) iodine/β-CD complex.

**Figure 2 pharmaceutics-13-00955-f002:**
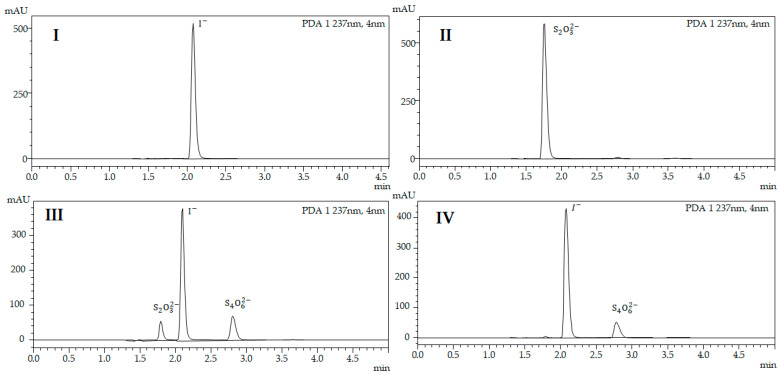
(**I**) Chromatogram of standard solution of potassium iodide; (**II**) Chromatogram of standard solution of sodium thiosulfate; (**III**) Chromatogram of iodine/β-CD complex with an excess amount of thiosulfate; (**IV**) Chromatogram of iodine/β-CD complex with an equivalent amount of thiosulfate.

**Figure 3 pharmaceutics-13-00955-f003:**
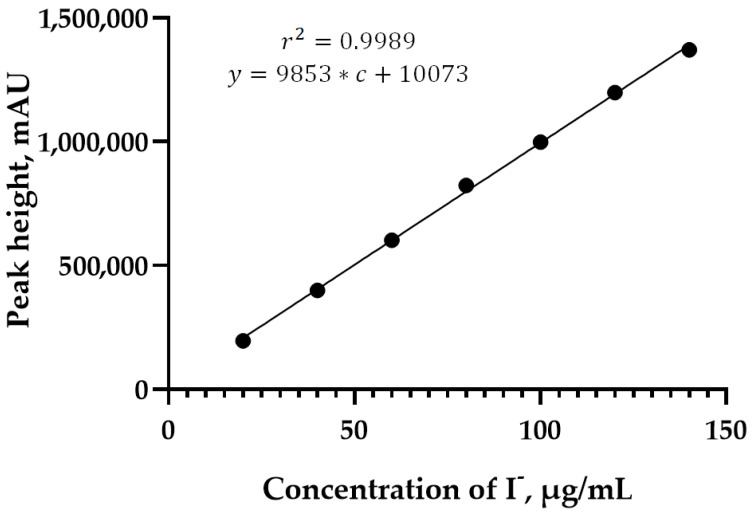
The dependence of peak height on the concentration of iodide.

**Figure 4 pharmaceutics-13-00955-f004:**
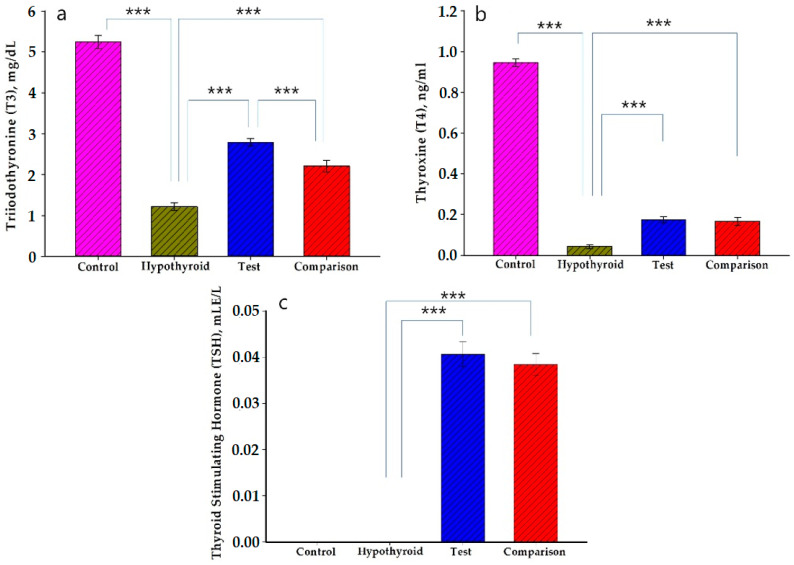
Results of Thyroid Stimulating Activity: (**a**) triiodothyronine (T3), (**b**) thyroxine (T4), (**c**) Thyroid-stimulating Hormone (TSH). Control, hypothyroid, test and comparison groups were given deionized water, methimazole, iodine/β-CD and potassium iodide, respectively (mean values ± SD). *** *p* < 0.001.

**Table 1 pharmaceutics-13-00955-t001:** Results of quantitation analysis.

No.	Sample Mass, g	Peak Height of Iodide in the Chromatograms, mAU		Amount of Iodine Species in Iodine/β-CD, % (*n* = 5)		RSD, %	
		Before Reduction	After Reduction	Before Reduction (Iodide Ions Only)	After Reduction (Total Iodine Species)	Before Reduction	After Reduction
1	0.0099	754,002 ± 6667	377,259 ± 5870	7.88 ± 0.07	19.91 ± 0.31	0.88	1.56
2	0.0099	756,404 ± 9077	376,903 ± 4453	8.03 ± 0.10	20.51 ± 0.24	1.20	1.18
3	0.0102	755,549 ± 7322	383,274 ± 5519	7.92 ± 0.08	19.61 ± 0.28	0.97	1.44
4	0.0105	755,656 ± 11,502	371,153 ± 7215	7.77 ± 0.12	19.46 ± 0.38	1.52	1.94
5	0.0098	755,650 ± 10,327	374,972 ± 4798	7.81 ± 0.11	20.07 ± 0.26	1.37	1.28

**Table 2 pharmaceutics-13-00955-t002:** The results of reduction yield.

No.	0.05 M Iodine Solution Volume, mL (K = 0.998)	Volume of Added 0.1 M Thiosulfate Solution, mL (K = 1.002)	Amount of Iodine, mg/mL		Yield, %	Average Yield, %
			Calculated	Found		
1	10.0 mL	10.0 mL	0.12675	0.1258	99.25	99.56 ± 0.27
2	10.0 mL	10.2 mL	0.12675	0.1263	99.64	
3	10.0 mL	10.5 mL	0.12675	0.1259	99.33	
4	10.0 mL	15.0 mL	0.12675	0.1264	99.72	
5	10.0 mL	20.0 mL	0.12675	0.2660	99.88	

**Table 3 pharmaceutics-13-00955-t003:** Results of linearity analysis.

No	Concentration of KI, µg/mL	Peak Height, mAU
1	20	196,075 ± 6766
2	40	399,471 ± 7397
3	60	602,044 ± 9254
4	80	822,947 ± 10,521
5	100	998,702 ± 16,791
6	120	1,198,334 ± 10,341
7	140	1,370,453 ± 12,195

**Table 4 pharmaceutics-13-00955-t004:** Results of validation parameters.

Parameters	Value
Linear Range (µg/mL)	20–140
$R^{2}$	0.9989
Accuracy (%)	98.41–102.87%
RSD (%) (overall)	1.33%
LOD	1.24
LOQ	4.14

## Data Availability

Data is contained within the article.

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
