# Peer review of "Development of an Improved Method for the Determination of Iodine/β-Cyclodextrin by Means of HPLC-UV: Validation and the Thyroid-Stimulating Activity Revealed by In Vivo Studies"

_pharmaceutics, 2021, doi:10.3390/pharmaceutics13070955_

Round 1
Reviewer 1 Report
Pharmaceutics_2021_1263302
The manuscript untitled “Development of an improved method for the determination of iodine\β-cyclodextrin by means of HPLC-UV: validation and the thyroid stimulating activity revealed by in vivo studies” by Avez Sharipov, Zufar Boboev, Sunnatullo Fazliev, Shokhid Gulyamov, Yunuskhodjayev Akhmatkhodja, Razzokov Jamoliddin submitted to section: Pharmaceutical Technology, Manufacturing and Devices.
The manuscript is a research article devoted to the iodine/β-cyclodextrin inclusion complex as a source of iodine, the use of the HPLC method for the quantitative analysis of iodine in the complex, and the assessment of thyrotropic activity of the complex.
Cyclodextrins are unique molecules for the creation of modern drugs based on inclusion complexes. They are also often considered as containers to improve the water solubility of drugs or delivery vehicles to target organs.
These materials will be useful to specialists in the field of biological chemistry and medicinal chemistry.
The manuscript does not raise any objections, and can be published in Pharmaceutics after minor revision.
Several details and inaccuracies should be noted. Authors should read their manuscript carefully and correct it.
- Abstract. “Our findings indicate higher thyroid stimulating activity of iodine/β-cyclodextrin compared to potassium iodide…” Based on the presented bioassay results, it seems premature to make such a conclusion.
- Introduction, lines 77-78. It would be good to compare the sizes of the internal cavities of α-CD and β-CD, and also compare them with the size of the iodine molecule in order to understand why, as it was found that β-CD forms a more stable complex with iodine than α-CD.
- Synthesis, line 107. “…38 g of I2 (1.5 mM) was dissolved in 15 mL of distilled water…” Speaking about the complexation and solubility of iodine in water, it should be noted that iodine does not dissolve in water.
- Line 114. The authors do not provide evidence of the composition of the studied complex.
- Lines 169-170. “Iodine/β-CD” was administered to rats in the test group at 1 mg/kg body weight daily for 10 days before an hour of the administration of MMI.” Why was the MMI (methimazole) drug administered to this group of animals?
- Lines 236-238, read “…and solutions resulted from reduction of the iodine/β-CD sample with equivalent (Figure 2, III) and excess amounts of thiosulfate (Figure 2, IV)”.
- Lines 242-243. Further, the caption to Fig. 2: “…(III) Chromatogram of iodine/β-CD complex with excess amount of thiosulfate; (IV) Chromatogram of iodine/β-CD complex with equivalent amount of thiosulfate.” Where is it right?
- Lines 251-252. “First, we determined amount of iodide in the complex.” Here's a mistake.
- Line 254. “…iodine/β-CD complex contains 7.88 ± 0.2 % iodide ions…” In Table 1, column “Before reduction (iodide ions only)” shows the value 88 ± 0.07 %. What deviation value should you accept?
- Section 3.5. “Results of Thyroid stimulating Activity and Discussions” it is necessary to add an explanation of why exactly the hormones T3, T4 and TSN were studied.
Author Response
|
Dr. Jamoliddin Razzokov Tashkent Institute of Irrigation and Agricultural Mechanization Engineers Kari Niyoziy Street 39, Tashkent 100000, Uzbekistan jamoliddin.razzokov@uantwerpen.be |
|
Pharmaceutics – MDPI
Prof. Dr. Yvonne Perrie Editor-in-chief
|
|
|
DATE 17 June 2021 |
SUBJECT Manuscript resubmission |
||
Dear Editor,
We would like to thank all referees for the evaluation of our manuscript submitted to Pharmaceutics (Manuscript ID: pharmaceutics-1263302) by Avez Sharipov, Zufar Boboev, Sunnatullo Fazliev, Shokhid Gulyamov, Yunuskhodjayev Akhmatkhodja, Razzokov Jamoliddin, entitled “Development of an improved method for the determination of iodine\β-cyclodextrin by means of HPLC-UV: validation and the thyroid stimulating activity revealed by in vivo studies”.
We read the referees’ reports carefully. All two referees recommended our paper for publication after considering some comments. Please find below our answers to the referees’ comments, as well as the corresponding changes and improvements made in the manuscript.
Sincerely,
Razzokov Jamoliddin
Reply to the Referees
We would like to thank the referees for the useful comments/suggestions that, indeed, are valuable and improve the quality of our manuscript. We have taken all comments into consideration and revised the manuscript accordingly. All additions and modifications to the manuscript are formatted in red text.
Reviewer #1:
The manuscript untitled “Development of an improved method for the determination of iodine\β-cyclodextrin by means of HPLC-UV: validation and the thyroid stimulating activity revealed by in vivo studies” by Avez Sharipov, Zufar Boboev, Sunnatullo Fazliev, Shokhid Gulyamov, Yunuskhodjayev Akhmatkhodja, Razzokov Jamoliddin submitted to section: Pharmaceutical Technology, Manufacturing and Devices.
The manuscript is a research article devoted to the iodine/β-cyclodextrin inclusion complex as a source of iodine, the use of the HPLC method for the quantitative analysis of iodine in the complex, and the assessment of thyrotropic activity of the complex.
Cyclodextrins are unique molecules for the creation of modern drugs based on inclusion complexes. They are also often considered as containers to improve the water solubility of drugs or delivery vehicles to target organs.
These materials will be useful to specialists in the field of biological chemistry and medicinal chemistry.
The manuscript does not raise any objections, and can be published in Pharmaceutics after minor revision.
Several details and inaccuracies should be noted. Authors should read their manuscript carefully and correct it.
- Abstract. “Our findings indicate higher thyroid stimulating activity of iodine/β-cyclodextrin compared to potassium iodide…” Based on the presented bioassay results, it seems premature to make such a conclusion.
Reply: We thank the referee for this comment. Indeed, we have mentioned the higher activity of iodine/β-cyclodextrin against the thyroid in comparison to potassium iodide in case of T3 hormone. According to Figure 3a, iodine/β-cyclodextrin showed significant increase in comparison with comparison group. According to references, we performed mostly used hypothyroidism model, as this model was more suitable in our work. In this respect, we modified the sentence accordingly:
Our findings indicate that thyroid stimulating activity of iodine/β-cyclodextrin is comparable to that of potassium iodide, which is the main active pharmaceutical substance of conventional drugs for iodine deficiency.
- Introduction, lines 77-78.It would be good to compare the sizes of the internal cavities of α-CD and β-CD, and also compare them with the size of the iodine molecule in order to understand why, as it was found that β-CD forms a more stable complex with iodine than α-CD.
Reply: The referee is right. Now, we have added more detailed information about the cavity size of α-CD and β-CD by explaining complex formation with iodide, see page 4 (Fig 1):
- The size of internal cavity of α-CD and β-CD is found to be 4.7-5.3Å and 6.0-6.5Å, respectively. The determined diameter of iodine molecules is 4.94 Å [16]. Thus, the stable complex formation of β-CD with iodine molecules is comparatively more probable than α-CD [16, 23].
- Synthesis, line 107. “…38 g of I2 (1.5 mM) was dissolved in 15 mL of distilled water…” Speaking about the complexation and solubility of iodine in water, it should be noted that iodine does not dissolve in water.
Reply: Indeed, our sentence has led to confusion. The sentence has been corrected as follows, see page 3:
- Briefly, 3.8 g of KI were dissolved in 15 mL of deionized water. Consequently, the prepared solution was mixed with 0.38 g of I2 (1.5 mM) and was left 10 minute to form polyiodide.
- Line 114. The authors do not provide evidence of the composition of the studied complex.
Reply: Indeed, we missed to mention and iodine/β-CD in the sentence. The sentence has been corrected accordingly, see page 3:
- The final product - iodine/β-CD complex was dried in vacuum for 4 h at 40 °C. Yield – 81.4%, humidity – 2.4%, purity – 97.32%.
- Lines 169-170.“Iodine/β-CD” was administered to rats in the test group at 1 mg/kg body weight daily for 10 days before an hour of the administration of MMI.” Why was the MMI (methimazole) drug administered to this group of animals?
Reply: We apologize for the lack of information. In the present study we induced the hypothyroidism by administration of methimazole (MMI) in the control group. Our aim was to check the thyroid-stimulating activity “Iodine/β-CD” by using this animal model, for this reason the test group was received MMI.
- Lines 236-238, read “…and solutions resulted from reduction of the iodine/β-CD sample with equivalent(Figure 2, III) and excess amounts of thiosulfate (Figure 2, IV)”.
Reply: The referee is right. We clarified the uncertainty in the sentence:
… and solutions resulted from reduction of the iodine/β-CD sample with excess (Figure 2, III) and equivalent amounts of thiosulfate (Figure 2, IV).
- Lines 242-243. Further, the caption to Fig. 2: “…(III) Chromatogram of iodine/β-CD complex with excessamount of thiosulfate; (IV) Chromatogram of iodine/β-CD complex with equivalent amount of thiosulfate.” Where is it right?
Reply: This sentence has been written correctly but we have made a mistake in sentence mentioned in 6th comment.
- Lines 251-252. “First, we determined amount of iodidein the complex.” Here's a mistake.
Reply:
- Line 254. “…iodine/β-CD complex contains 7.88 ± 0.2 % iodide ions…” In Table 1, column “Before reduction (iodide ions only)” shows the value 88 ± 0.07 %. What deviation value should you accept?
Reply: We admit this has led to confusion. The value (7.88±0.2%) given in the line 254 is average of all 25 experiments (for each such value given in Table 1, 5 independent experiments were performed) given in Table 1, column 5. The value given in the first row (7.88±0.07%) of Table 1 is the result of 5 independent experiments only. We now mentioned n=25 and n=5 in the main text and in Table 1, respectively.
- Section 3.5. “Results of Thyroid stimulating Activity and Discussions” it is necessary to add an explanation of why exactly the hormones T3, T4 and TSN were studied.
Reply: We apologize for our lack of clarity. In the present study T3, T4 and TSH serum levels were measured in animal groups. The reason why these hormones were chosen is that the thyroid gland releases the two main thyroid hormones, triiodothyronine (T3), thyroxine (T4) and TSH stimulates thyroid follicular cells to release thyroid hormones in the form of T3 or T4. T3, is the active form of thyroid hormone. Tetraiodothyronine, also known as thyroxine or T4, equals more than 80% of the secreted hormone. As known, when low levels of TSH, lack of stimulation of thyroid follicular cells causes T3 and T4 levels reduction, thus hypothyroidism. It is found that the T3, T4 and TSH hormones are one of the main biomarkers in the hypothyroidism. We have now revised as following:
The thyroid gland releases the two main thyroid hormones, triiodothyronine (T3), thyroxine (T4). T3, is the active form of thyroid hormone and T4, equals more than 80% of the secreted hormone. TSH stimulates thyroid follicular cells to release thyroid hormones in the form of T3 or T4. It is well known, when low levels of TSH, lack of stimulation of thyroid follicular cells causes T3 and T4 levels reduction, which leads to hypothyroidism. From this reason we chose these hormones.

Reviewer 2 Report
Authors described the development of a new method for the quantification of iodine in ß-CD complexes by HPLC and the study of the effects on an animal model.
I think their XRD-analysis is a good point to start the characterization of the complex formation and their results are consistent. The HPLC method is robust, and the reduction of the species seems reasonable to reach higher quantity in the CD. I do not if is feasible for them, but I miss a release study in vitro of the iodine from the CD complexes, just to ensure the right quantity of iodine is delivered.
In line 107, I do not really understand why they use water to solubilize the iodine if is not water soluble.
They have an interesting experiment with animals as model of disease and the effect of the formulations in the serum levels of hormones. Nevertheless, I miss some comparisons/liaison between the quantification and the results in the animals, perhaps to establish a relationship between the release kinetic of the drug in vitro and in vivo. I think this will be an added value to their results, even to explain the required time to reach those effects. Some references to compare their results with previous publication would be helpful too.
In the last section, I think it is necessary to explain why they choose to analyze the levels of those hormones in the animals. A brief explain in the beginning of the section could help the readers.
I encourage the authors to do not use contractions in the manuscript , as in line 246 “…don´t…”.
Author Response
|
Dr. Jamoliddin Razzokov Tashkent Institute of Irrigation and Agricultural Mechanization Engineers Kari Niyoziy Street 39, Tashkent 100000, Uzbekistan jamoliddin.razzokov@uantwerpen.be |
|
Pharmaceutics – MDPI
Prof. Dr. Yvonne Perrie Editor-in-chief
|
|
|
DATE 17 June 2021 |
SUBJECT Manuscript resubmission |
||
Dear Editor,
We would like to thank all referees for the evaluation of our manuscript submitted to Pharmaceutics (Manuscript ID: pharmaceutics-1263302) by Avez Sharipov, Zufar Boboev, Sunnatullo Fazliev, Shokhid Gulyamov, Yunuskhodjayev Akhmatkhodja, Razzokov Jamoliddin, entitled “Development of an improved method for the determination of iodine\β-cyclodextrin by means of HPLC-UV: validation and the thyroid stimulating activity revealed by in vivo studies”.
We read the referees’ reports carefully. All two referees recommended our paper for publication after considering some comments. Please find below our answers to the referees’ comments, as well as the corresponding changes and improvements made in the manuscript.
Sincerely,
Razzokov Jamoliddin
Reply to the Referees
We would like to thank the referees for the useful comments/suggestions that, indeed, are valuable and improve the quality of our manuscript. We have taken all comments into consideration and revised the manuscript accordingly. All additions and modifications to the manuscript are formatted in red text.
Reviewer #2:
Authors described the development of a new method for the quantification of iodine in ß-CD complexes by HPLC and the study of the effects on an animal model.
- I think their XRD-analysis is a good point to start the characterization of the complex formation and their results are consistent. The HPLC method is robust, and the reduction of the species seems reasonable to reach higher quantity in the CD. I do not if is feasible for them, but I miss a release study in vitro of the iodine from the CD complexes, just to ensure the right quantity of iodine is delivered.
Reply: We thank the referee for this comment. Since here we’re only concerned with development of quantification method, we did not present results of pharmacokinetics studies. Because it’s still ongoing project. For the analytical purpose however release of iodine does not affect the procedure, because the reduction reaction with thiosulfate is very fast and quantitative. Detailed pharmacokinetics study will be published elsewhere.
- In line 107, I do not really understand why they use water to solubilize the iodine if is not water soluble.
Reply: Indeed, our sentence has led to confusion. The sentence has been corrected as follows, see page 3:
- Briefly, 3.8 g of KI were dissolved in 15 mL of deionized water. Consequently, the prepared solution was mixed with 0.38 g of I2 (1.5 mM) and was left 10 minute to form polyiodide.
- They have an interesting experiment with animals as model of disease and the effect of the formulations in the serum levels of hormones. Nevertheless, I miss some comparisons/liaison between the quantification and the results in the animals, perhaps to establish a relationship between the release kinetic of the drug in vitro and in vivo. I think this will be an added value to their results, even to explain the required time to reach those effects. Some references to compare their results with previous publication would be helpful too.
Reply: We thank the referee for this comment. Since here we are only concerned with development of quantification method, we did not present results of pharmacokinetics studies. Because it is still ongoing project. Detailed in vitro and in vivo pharmacokinetics study will be published elsewhere.
- In the last section, I think it is necessary to explain why they choose to analyze the levels of those hormones in the animals. A brief explain in the beginning of the section could help the readers.
Reply: We agree with this and have incorporated your suggestion throughout the Results and Discussion section. In the present study T3, T4 and TSH serum levels were measured in animal groups. The reason why these hormones were chosen is that the thyroid gland releases the two main thyroid hormones, triiodothyronine (T3), thyroxine (T4) and TSH stimulates thyroid follicular cells to release thyroid hormones in the form of T3 or T4. T3, is the active form of thyroid hormone. Tetraiodothyronine, also known as thyroxine or T4, equals more than 80% of the secreted hormone. As known, when low levels of TSH, lack of stimulation of thyroid follicular cells causes T3 and T4 levels reduction, thus hypothyroidism. It is found that the T3, T4 and TSH hormones are one of the main biomarkers in the hypothyroidism. We have now revised as following:
The thyroid gland releases the two main thyroid hormones, triiodothyronine (T3), thyroxine (T4). T3, is the active form of thyroid hormone and T4, equals more than 80% of the secreted hormone. TSH stimulates thyroid follicular cells to release thyroid hormones in the form of T3 or T4. It is well known, when low levels of TSH, lack of stimulation of thyroid follicular cells causes T3 and T4 levels reduction, which leads to hypothyroidism. From this reason we chose these hormones.
- I encourage the authors to do not use contractions in the manuscript , as in line 246 “…don´t…”.
Reply: We thank the referee for this comment. Found contractions were corrected.

Round 2
Reviewer 2 Report
Authors addressed correctly all the comments made. They are going to publish in the future the results regarding the pharmacokinetics. I am agree to publish this manuscript in the current form.
Author Response
Reply to the Referees
We would like to thank the referees for the useful comments/suggestions that, indeed, are valuable and improve the quality of our manuscript. We have taken all comments into consideration and revised the manuscript accordingly. All additions and modifications to the manuscript are formatted in red text.
Reviewer #2:
Authors described the development of a new method for the quantification of iodine in ß-CD complexes by HPLC and the study of the effects on an animal model.
- I think their XRD-analysis is a good point to start the characterization of the complex formation and their results are consistent. The HPLC method is robust, and the reduction of the species seems reasonable to reach higher quantity in the CD. I do not if is feasible for them, but I miss a release study in vitro of the iodine from the CD complexes, just to ensure the right quantity of iodine is delivered.
Reply: We thank the referee for this comment. Since here we’re only concerned with development of quantification method, we did not present results of pharmacokinetics studies. Because it’s still ongoing project. For the analytical purpose however release of iodine does not affect the procedure, because the reduction reaction with thiosulfate is very fast and quantitative. Detailed pharmacokinetics study will be published elsewhere.
- In line 107, I do not really understand why they use water to solubilize the iodine if is not water soluble.
Reply: Indeed, our sentence has led to confusion. The sentence has been corrected as follows, see page 3:
- Briefly, 3.8 g of KI were dissolved in 15 mL of deionized water. Consequently, the prepared solution was mixed with 0.38 g of I2 (1.5 mM) and was left 10 minute to form polyiodide.
- They have an interesting experiment with animals as model of disease and the effect of the formulations in the serum levels of hormones. Nevertheless, I miss some comparisons/liaison between the quantification and the results in the animals, perhaps to establish a relationship between the release kinetic of the drug in vitro and in vivo. I think this will be an added value to their results, even to explain the required time to reach those effects. Some references to compare their results with previous publication would be helpful too.
Reply: We thank the referee for this comment. Since here we are only concerned with development of quantification method, we did not present results of pharmacokinetics studies. Because it is still ongoing project. Detailed in vitro and in vivo pharmacokinetics study will be published elsewhere.
- In the last section, I think it is necessary to explain why they choose to analyze the levels of those hormones in the animals. A brief explain in the beginning of the section could help the readers.
Reply: We agree with this and have incorporated your suggestion throughout the Results and Discussion section. In the present study T3, T4 and TSH serum levels were measured in animal groups. The reason why these hormones were chosen is that the thyroid gland releases the two main thyroid hormones, triiodothyronine (T3), thyroxine (T4) and TSH stimulates thyroid follicular cells to release thyroid hormones in the form of T3 or T4. T3, is the active form of thyroid hormone. Tetraiodothyronine, also known as thyroxine or T4, equals more than 80% of the secreted hormone. As known, when low levels of TSH, lack of stimulation of thyroid follicular cells causes T3 and T4 levels reduction, thus hypothyroidism. It is found that the T3, T4 and TSH hormones are one of the main biomarkers in the hypothyroidism. We have now revised as following:
The thyroid gland releases the two main thyroid hormones, triiodothyronine (T3), thyroxine (T4). T3, is the active form of thyroid hormone and T4, equals more than 80% of the secreted hormone. TSH stimulates thyroid follicular cells to release thyroid hormones in the form of T3 or T4. It is well known, when low levels of TSH, lack of stimulation of thyroid follicular cells causes T3 and T4 levels reduction, which leads to hypothyroidism. From this reason we chose these hormones.
- I encourage the authors to do not use contractions in the manuscript , as in line 246 “…don´t…”.
Reply: We thank the referee for this comment. Found contractions were corrected.
